# Elements of Divergence in Urbanization between Central and Eastern Europe (CEE) and the Core of the Continent

**Péter Faragó** [1,*], **Krisztina Gálos** [2] **and Dávid Fekete** [3]

1 Doctoral School of Regional and Business Administration Sciences, Széchenyi István University, 9026 Győr, Hungary
2 Kautz Gyula Faculty of Economics, Széchenyi István University, 9026 Győr, Hungary
3 Department for Regional Sciences and Public Policy, Széchenyi István University, 9026 Győr, Hungary
* Correspondence: farago.peter@sze.hu

**Abstract:** There is an ongoing debate regarding whether the EU-10 converges at the core of Europe or not. Although the evidence supports both perspectives, the gap in urbanization is undeniable. In this explorative study, two economic processes contributing to this disparity—foreign direct investment and migration—were analyzed and contextualized with respect to urbanization using grounded theory. It was concluded that there is slight convergence in the frontier, usually in urbanized areas of Central and Eastern Europe (CEE), but not in the rural areas; additionally, the rural–urban dichotomy within the CEE countries is deepening due to the self-enhancing nature of the analyzed processes.

**Keywords:** urbanization; Central and Eastern Europe; FDI; migration; concentration; divergence



## 1. Introduction

Thirty years ago, a wave of new opportunities opened up for socialist states. The transition was meant to entail urban transformation. The available knowledge of urbanization at the core of Europe (hereafter referred to as "core") is extensive due to the long history of scientific research in various related areas. However, as far as Central and Eastern Europe (CEE; see definition in Section 2) and the Baltics are concerned, in-depth qualitative knowledge is scarce, proven by the fact that the aimed and anticipated economic convergence between the two sides of Europe is ambiguous. Components of this phenomenon (or its absence) are urbanization and the rural–urban dichotomy [1].

The political and economic vacuums that were created after the fall of the USSR were filled by Western policies and capital aiming to reshape the CEE countries. Accordingly, the balance-oriented socialist approach was substituted with competition, and the emerging unequal growth rates were ignored. Alongside the adoption of capitalist mechanisms, the management of urban spaces became decentralized and more fragmented [2], opening up a path for unmoderated divergence.

Regarding CEE, studies have emphasized the path dependencies of administrative and economic patterns, particularly the (still) centralized configurations, insufficient administrative capacities and adaptability [3], and the fact that foreign direct investment (FDI), which is defined as the inflow of capital possessed and directed by foreigners, as a new dynamism, prefers the well-established industrial hubs (i.e., traditional structures) that were inherently more developed under socialism [4]. The infrastructural environment of CEE is also poor compared to the core of Europe; ruins or old buildings that are in a state of disrepair can be seen in Southern Hungary, Eastern Poland and Bulgaria because reconstruction fell behind in those regions, whereas in the core, even the villages have recovered since the war. Schmidt [5] exemplified this landscape by comparing former Federal Germany with Democratic Germany. Infrastructural recovery, especially in a technologically underdeveloped environment, requires a substantial labor force, but the decline in the native population, including through emigration (except in the capital regions of

CEE [6]), has hindered such efforts. Based on the latest Eurostat data, a low and declining fertility rate (1.5 in 2020) has been observed throughout the entire continent.

In this explorative study, two economic processes prevalent in CEE—FDI and labor migration—were assessed and contrasted with the rest of the continent. We aimed to argue why FDI and labor migration flow are relevant to urbanization and rural–urban disparities and divergences across Europe and in domestic frames (rural–urban areas). We believe that emphasizing the correspondence regarding these economic tendencies, urbanization and the resulting gaps (in performance and perhaps welfare too) between rural and urban areas and the entire continent prompts policymakers to develop some countermeasures aimed at social equality and sustainability.

## 2. Materials and Methods

This study followed an explorative approach, focusing on CEE's uniqueness in contrast with the core. Based on the World Bank's [7] terminology, CEE or the EU-10 region covers the following states: Bulgaria, the Czech Republic, Estonia, Hungary, Lithuania, Latvia, Poland, Romania, Slovenia and Slovakia. Based on a literature review and additional informal sources, FDI, labor migration and their impacts on CEE were discovered in this study using grounded theory (GT). Occasionally, we also shared personal observations and experiences (informal sources).

We analyzed foreign capital investments and labor migration as processes that persistently allowed urban divergence. When the CEE countries gained access to the European Union's (EU's) common market, the barriers to the free flow of labor force, capital, money and commodities were (gradually) dismantled. Advocates of neoliberal economics argued that these steps and launched processes would eventually culminate in an economic and social equilibrium via regional integration within the continent, but this has not happened. In this paper, two of the four factors have been scrutinized as catalysts for the opposite reality. A relatively higher volume of FDI's and labor movements was considered as a threshold for selection, while the financial and commodity markets are less stable and more flexible, swiftly adaptable in a sense, although their interplay with urbanization can be a subject of investigation.

GT is a systematic approach involving simultaneous data collection and conceptualization and was originally described by Glaser and Strauss in the book *Discovery of GT U* [8]. GT users mainly deal with qualitative knowledge, and novel conclusions may be drawn from multiple disciplines [9]. Data, theories and statistics are flexibly contextualized and synthesized by relying on the researcher's creativity. As we researched the literature, we became able to identify the fundamental tendencies concerning the core of GT. We complimented the past findings with our personal observations, experiences and interviews, that is, with information from informal and non-scientific sources. According to the GT principles, reviewing the scientific literature in advance would "contaminate" the results, blinding us to new perspectives [10]. However, we believe that some initial knowledge was desirable to ascertain what was worth looking into, as well as which questions were relevant, and whether there were any gaps, discrepancies or contradictions between intentions/plans and reality that needed to be clarified and comprehended.

Urbanization is a dynamic process. GT was deemed suitable for researching urbanization because no general pattern is applicable for any country that undergoes urbanization; moreover, it is hard to identify which contextual elements play a role and to what extent, per location (e.g., as described in [11]). We focused on comparing two areas in terms of two specific characteristics—FDI and labor migration—due to space limitations. Characteristics of the two contrasting geographical areas (e.g., differences between the Baltics and the V4 countries) have not been covered in this paper.

## 3. Theoretical Background: Convergence or Divergence?

Researchers have found that the catch-up and convergence of economies have remained unapproved or incomplete ever since 1990 (e.g., with respect to rates of poverty

or social exclusion [12]). However, controversial evidence has also been provided: the presence of the CEE economies in the international market has substantially increased [13], a tendency capacitated by FDI and economies of scale. Nevertheless, quality-related issues still weigh heavy. While the region's value addition is dynamically increasing, it remains relatively low and spatially quite unequal. Neither the composition nor destination of export is appropriately diversified [14]. Conversely, the service sector—characterized by high value addition—is developing in CEE, although its share in the region's gross domestic product (GDP) is still below the respective level of the core [13]. Some convergence can be seen at the macroeconomic level, but a closer look reveals that development is generally delivered by large, industrialized cities, while the countryside remains neglected. This gap is broadened by urbanization.

Considering the definition of urbanization as "an increase in the number of people living and working in a city or metropolitan area" [15], Figure 1 (Source: World Bank Data, own editing) demonstrates the difference in urbanization between CEE and the rest of Europe.

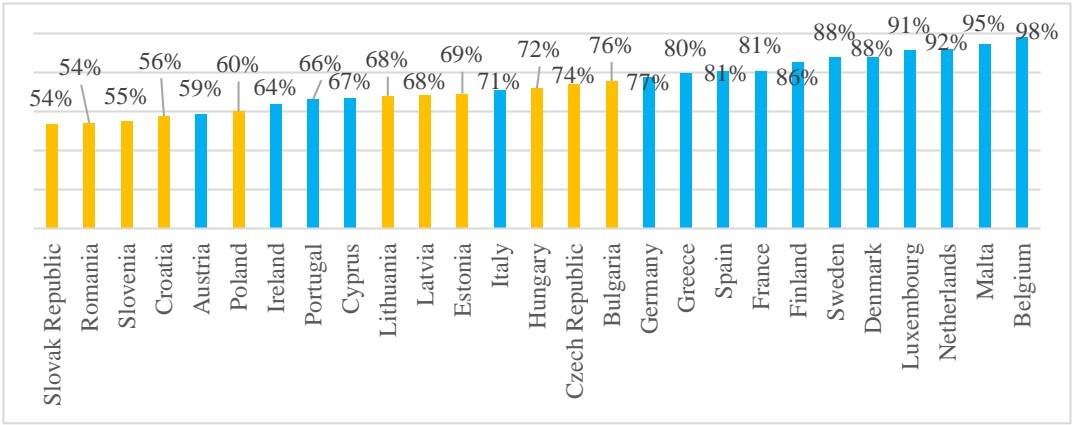

**Figure 1.** Share of population living in urban areas across Europe in 2020; Central and Eastern European countries are denoted by yellow.

Defining areas as urban and rural is hardly feasible methodologically. Depending on the country, the demarcation between the two categories is based on different requirements (e.g., population, provision of infrastructure, availability of services, presence of public institutions). Moreover, this categorization is static; therefore, we deem it to be more valid if we talk about urbanization as a dynamic process and use the terms "urban" and "rural" not in accordance to a location-biased definition [16].

"It is an established fact that urbanization in developed countries is accompanied by economic growth and industrialization which mutually self-reinforce one another" [17]. Particular economic structures (employment, value addition and output per sector) are closely associated with the urban–rural dichotomy/divergence. For example, primary economic branches (e.g., agriculture) and some secondary ones (e.g., textile, smelting) usually reside in rural areas, not in metropolitan centers. The majority of the CEE population is urban-dwelling, but its share is far below the EU's average (75% in 2020 [18]). Conclusively, the dispersion of the population also denies the completion of convergence between the old and new EU states. In agreement with past researchers [12,19,20], we acknowledge that the gaps within the individual CEE countries are also significant, but investigating these was beyond the scope of the present study.

Driven by the fact that urbanization is reflected in economic structures and, thus, development, we proposed the following question for this study, in agreement with the convergence-denying argument: How and why did FDI and labor migration reinforce rural–urban disparities (1) between the core of Europe in CEE and (2) within the individual CEE countries?

There is substantial literature on the characteristics that differentiate CEE from the core: geographical size, exceeding dependency on one economic sector and its global value chain, higher vulnerability to capitalism [21], risk of poverty and material deprivation rate predesting social exclusion [12], inefficient use of space and functionality [2], population densities and dispersion [19], comparative advantages, reliance on external finance and capital [13], investment into human capital [22], exposure to EU funding, etc. Based on the assumption of Europe-wide variations, it is reasonable to claim that the evolution of urbanization is location-specific.

## 4. Analyzing How FDI and Labor Migration Have Been Facilitating Urban Transformation in CEE

*4.1. Transition from State-Led Socialism to Capitalism and Liberalism*

Presently, population proportions as well as the economic performances divide metropolitan areas from the countryside and Eastern Europe from the core countries of the continent. Low degrees of urbanization can be explained by the correlation of low compactness (less function with the same amount of space) with higher dispersion [2]. This means that economic activities are arranged in a more concentrated manner in the core, making cities "concise" and densely populated, thus generating high outputs on a high share of build space. Spectacular proof is in the frequency of taller buildings in core cities. This does not mean, however, that cities in the core have a smaller horizontal extent. Due to the earlier start of urbanization in the core, the proportion of urban areas with respect to total land is approximately twice that of CEE, which can be observed in Figure 1. Extremely compact/utilized settlements are mainly found in the core, whereas to the east, beyond CEE, Moscow is the only city with a scale comparable to that of Brussels, for example. Cities and their urban functions are generally more dispersed in CEE, as urban lands are less built upon/utilized. As Taubenböck [2] emphasized, the 25-year delay in the urbanization process has left its mark.

As Wolff [19] argued, due to the abrupt urbanization after 1990, higher population densities occur in CEE cities only sporadically. In contrast, the subsequent stages of urbanization (de-densification and suburbanization) were more advanced in the core, suggesting a philosophical shift in preferences: Growth is the superior goal in CEE, while being calm, green and a "family-friendly" environment, etc., are favored in the core, even at the price of longer commute times to the city centers and higher expenses. Daily commuting presupposes the availability of better road networks and flexible alternative transportation methodologies (e.g., car-sharing) as well as car ownership, that ease the burden of traveling.

Since the 1970s, the CEE cities have experienced rapid growth. The oil crisis made socialist rulers realize that the global surge in fossil fuel prices could not be balanced out arbitrarily; external funding was necessary. Accumulated debt increased the central administration's perception of the current system as unsuitable and unsustainable [23]. Due to the inevitable shift toward a liberalized free market, as Taubenböck [2] explained, cities started to grow rather suddenly, concentrating their resources to harvest higher efficiency and becoming more compact; respective dynamisms were the most intensive in Warsaw and Moscow. This transformation could be exclusively seen in the CEE capitals, with these regions displaying the typical aspects of capitalist development trajectories.

The neoliberal balance-claiming theory could not explain the deepening economic divide and uneven development across Europe. Privatization, liberalization and free-market competition were among the goals for the economic and political transition, but the democratic capitalist principles were not consensually accepted in CEE, as opposed to the core [24]. According to Wike [24], something that distinguishes CEE from the rest of the continent is the stronger or weaker skepticism of some of these countries' citizens (e.g., Hungary and Lithuania) toward the Western alliance systems and new political course alongside praises for the past socialist era when life used to be "predictable and plannable," as Tölgyessy [23] had proclaimed. The political transition resulted in instability, economic pressure and hardships: Together, these made people perceive capitalism, liberalism and

the "imported" democratic institutions as being unable to provide prosperity and welfare. This sentiment was bolstered by the skyrocketing unemployment and inflation rates, direct drop in living standards and increase in illegitimate forms of privatization [25]. These outcomes eliminated the initial enthusiasm.

Socially and legally, the transition could be captured through the inflow of Western laws and practices, but informal networks and unofficial power centers latently persisted [26]; in other words, the persons behind the old and new offices often remained the same. Consequently, institutions in CEE were perceived as powerless or biased and were bestowed with less significance and trust [25]. The perceived powerlessness of local decision makers and suspicious attitudes toward liberalism and capitalism have contributed to the comprehensive, long-term passivity and restraint of the community. Manifestations of passivity are absent partnership-seeking, reciprocity, initiation, confidence, trust, engagement and risk-taking. Therefore, entrepreneurial characteristics [27] are less mature in CEE than in the core, which is reflected in the dependency on foreign capital and know-how inflow, as detailed below.

Economically, since the 1980s, growth-oriented principles, integration first into the European and then into the global markets, and industrialization have enrooted divergence among capital cities, where the associated effects are the strongest; conversely, smaller settlements remain confined. Major variations have emerged in terms of pace, depth, resistance and commitment surrounding the reforms of municipalities and competent local authorities [28]. The restructuring and adjustment to the global economy's demand entailed the fading of the national boundaries and dismantling of the domestic production networks (resulting in socialism), which were meant to meet only a limited local demand with different criteria regarding quality, for instance. Joining the international network of cities and gradually enhancing these linkages were ambitioned, but these were achieved to various extents by various settlements. As Milanovic concluded [28], the network of capital cities underwent and still is undergoing this integration dynamically, while towns and rural areas are struggling to gain international status and competitiveness.

Concluding from the articles of Milanovic [28] and Taubenböck [2], the major milestones of the urban transformations in the aftermath of the transition are drafted in Table 1. In parallel, the legislative and administrative progress is presented on the right-hand side [29].

**Table 1.** Urban and economic processes after the transition and their facilitator legal documents and administrative measures.

| Phase | Approximate Period | Fundamental Shift | Legal Premises |
|:---:|:---:|:---:|:---:|
| 1 | Right after the legal change in the regime | Abruptly replaced, centrally organized and hierarchical socialist structure | Single Market Accession (1992) |
| | | | Copenhagen Summit (1993) |
| | | | Agenda 2000 |
| | | | CEE EU accession (2004) |
| 2 | During intensifying internationalization in the 1990s, enhanced by the accession negotiations | Faster or sluggish privatization, regeneration and integration to the globally stretching supply chains started | Europe 2000 (1991); Europe 2000+ (1994) |
| | | | European Spatial Development Perspectives (1999) |
| | | | Report on Economic and Spatial Cohesion (2001, 2004) |
| 3 | After adopting the European Union's administrative frames | On domestic scopes, particular cities emerged, taking the leading position in the process. | European Spatial Planning Observatory Network (ESPON) |

Despite the implementation of legislative and top-down planned measures aimed at facilitating a sustainable and equitable development trajectory, the following issue arises from the actual immature and imbalanced urban transformation: urban structures and the

population are dependent on external funding [13]. Notably, many rural CEE territories are still underdeveloped in many respects: infrastructure, innovation potential, modern and supportive institutional background, a culture of trust, undiversified economic structure, etc. [25]. The aftermath of the transition saw an acceleration in urbanization, although it was never at the same stage in CEE as in the core [19]. Polarization happened quite suddenly in CEE, while it was a more gradual process in Western Europe [2]. However, although the core economies have higher value addition on average, coupled with a more diversified structure, domestic imbalances are sometimes deeper there (the UK, France, Italy) than in CEE. Therefore, it can be concluded that neither high value addition nor diversified economic structures necessarily lead to balanced spatial development, as opposed to polarization [30]. Disparities in the core certify Kata's [31] conclusion and, accordingly, the concentration of production factors and evident economic and income growth facilitate polarization, not equilibrium or social sustainability.

As a personal input, we would like to highlight a crucial issue that has been a hot topic since the Russia–Ukraine war started in 2014 and has escalated recently, in 2022. The Hungarian MOL Group covers and supplies numerous countries in the region by refining Russian fossil fuels. Russian imports are under EU embargo nowadays (Summer, 2022), and, therefore, the energy prices have soared. The (fixed) asset portfolio of the legal predecessor of MOL was essentially designed for refining the Russian type of energy carriers (Urals oil). In other words, in the short run, Hungary is incapable of diversifying its energy supplier base due to this particular path dependency deriving from the socialist era. In an interview, Hungary's Minister of Foreign Affairs and Trade, Peter Szijjarto, stated, "Overhauling Hungary's oil infrastructure to handle crude from elsewhere could cost up to €18bn" [32]. Contrarily, principles of liberalism and capitalism hypothesize a perfect responsiveness and immediate adaptability from each market player without such constraints as exemplified by the Ural oil exposure of MOL.

*4.2. FDI: Specialization and Concentration*

Foreign capital inflow, especially direct investments, has played a peculiar role in CEE over the past decades from several angles. Since savings of the population have stagnated and local currencies have depreciated, foreign investments have been the virtual pillars of financial recovery and credit boom to enable the consolidation of the domestic capital accounts after the economic transition and downgrade. The same can be said about the Great Recession as well. Further, intangible assets were also delivered as a result, such as managerial skills or patents [33]. The fact that the foreign capital inflow exceeded domestic surpluses was the reason for us focusing on the occupational areas of FDI and how it affected urbanization.

During the socialist era, citizens and workers were centrally and effectively organized, but this came to an end. The newly arrived foreign capital has supplied no managerial and organizational knowledge, which are indispensable tools for modernizing local economies; moreover, it has also resulted in one-sided dependency. While employment was indeed boosted, phases of value chains with high value addition and profitability, such as R&D, were restricted to the origin countries. Pellényi [14] compared the employment distribution rates of some CEE economies and three advanced economies, and the findings are listed in Table 2 (Source: [14], own editing). The year 2011 was almost a decade after the EU accession of the listed countries, yet the indicators presented a significant spread, especially of fabrication. Except for fabrication, all other types of occupation were functionally upgraded and characterized by high value addition. Values higher than the average are colored in Table 2; orange represents disadvantageous subjects of specialization; green marks advantageous ones. Highlighting different dates was meant to reflect on the tendency of specialization.

**Table 2.** Manufacturing employment shares by occupation type (%) in 2011 and 2018.

| | 2011 | | | | 2018 | | | |
|---|---|---|---|---|---|---|---|---|
| Countries | R&D | Fabrication, Assembly | Business Support | Distribution and Sales | R&D | Fabrication, Assembly | Business Support | Distribution and Sales |
| The Czech Republic | 2.1 | 77.5 | 18.2 | 2.2 | 4.3 | 74.4 | 19.5 | 1.8 |
| Hungary | 5.1 | 78.1 | 14.8 | 2.1 | 6.6 | 74.1 | 16.2 | 3.0 |
| Poland | 4.5 | 74.2 | 19.9 | 1.5 | 5.3 | 72.6 | 20.6 | 1.6 |
| Romania | 6.4 | 80.7 | 10.8 | 2.1 | 7.6 | 81.8 | 8.8 | 1.9 |
| Slovakia | 2.2 | 78.6 | 17.6 | 1.6 | 2.3 | 77.1 | 17.9 | 2.7 |
| Finland | 14.3 | 63.3 | 20.8 | 1.6 | 17.0 | 60.3 | 20.9 | 1.9 |
| Germany | 8.6 | 63.0 | 22.9 | 5.5 | 9.1 | 59.0 | 26.0 | 5.9 |
| Sweden | 7.2 | 67.5 | 21.9 | 3.4 | 10.2 | 58.8 | 28.2 | 2.8 |
| **Average** | **6.3** | **72.9** | **18.4** | **2.5** | **7.8** | **69.8** | **19.8** | **2.7** |
| **Spread** | **4.0** | **7.2** | **4.0** | **1.4** | **4.5** | **9.0** | **6.0** | **1.4** |

Table 2 shows that the CEE countries persistently specialized in fabrication. The increased spread value from 2011 to 2018 denies convergence and the shift to sophisticated activities. Conclusively, the means of participation in the global value chains became more imbalanced among the analyzed countries in the given period. Fabrication and assembly lines require large industrial areas; therefore, they are rarely deployed in densely populated cities and are usually found in rural settlements as greenfield investments. The opposite is true for state-of-the-art research [34] or distribution centers, such as ports [35].

FDI determines which technologies and fields are to be prioritized, and its influence in CEE proved to be higher during the evolution of regional economics [13] than in the core. There are numerous examples of the CEE cities in which a single (foreign) enterprise is the paramount pillar of the local economy—Panasonic in Pardubice, Kia in Komárom, Audi in Győr, Hanza in Wroclaw, etc.—and its industries are inevitably prioritized. The influence of a foreign business entity or industry tends to be proportionately smaller and less frequent in Western European settlements [13], because they traditionally host diverse investments, and their domestically owned value chains are well established. Due to this, local economies in CEE are more exposed to foreign capital. Apart from occupational stratification, we also have to recall geographical concentration as another realm of unbalance.

As an example of the geographical FDI inflow concentration, Table 3, based on the data of the Hungarian Statistical Office [36], illustrates how regions of Hungarian Nomenclature of Territorial Units for Statistics (NUTS), level 3, benefitted or lost foreign investments. Additionally, coloration highlights absolute values exceeding 50%. It is clearly visible that the capital (also including the county) attracted by far the highest FDI, while Central Transdanubia was the least credited area. The most important message here is that regions had stable positions on balance: They were either continuous capital receivers or losers, more or less, throughout the past decade. FDI concentrated persistently to particular NUTS 3 regions (county seats of Central and Western Transdanubia) and withdrew from others, despite the fact that the production factors might have been cheaper there (rural areas of Transdanubia and Northern Hungary). No major changes occurred in their popularity from 2009 to 2020. Owing to space limitations, we refrained from presenting more countries here.

**Table 3.** FDI inflow into Hungarian NUTS level 3 regions (1000 EUR) and the share (%) of the individual regions in the total annual inflow between 2009 and 2020.

| Name of Territorial Units | 2009 | | 2010 | | 2011 | | 2012 | | 2013 | | 2014 | | 2015 | | 2016 | | 2017 | | 2018 | | 2019 | | 2020 | |
|---|---|---|---|---|---|---|---|---|---|---|---|---|---|---|---|---|---|---|---|---|---|---|---|---|
| Budapest | 5,424,956 | 106% | 495,362 | 51% | −830,773 | −21% | 3,182,372 | 52% | 582,907 | 49% | 2,086,943 | 34% | −3,947,353 | 122% | −2,831,971 | 261% | −1,211,774 | 146% | 3,421,854 | 72% | 1,641,202 | 35% | 1,748,441 | 25% |
| Pest | 248,043 | 5% | −247,993 | −25% | 181,391 | 4% | −2,098 | 0% | 276,068 | 23% | −504,882 | −8% | 81,392 | −3% | −45,436 | 4% | 123,124 | −15% | 358,254 | 8% | −58,798 | −1% | 1,261,463 | 18% |
| Central Transdanubia | −170,578 | −3% | 101,139 | 10% | 384,979 | 10% | 561,807 | 9% | 227,867 | 19% | 539,204 | 9% | 543,963 | −17% | 689,283 | −63% | 1,505,048 | −181% | 443,023 | 9% | 1,246,316 | 27% | 636,041 | 9% |
| Western Transdanubia | −655,342 | −13% | 3738 | 0% | 3,399,630 | 84% | 2,703,199 | 44% | 676,900 | 57% | 867,020 | 14% | −330,025 | 10% | −483,115 | 44% | −2,289,530 | 275% | −1,189,470 | −25% | 2242 | 0% | −280,159 | −4% |
| Southern Transdanubia | −59,902 | −1% | −23,074 | −2% | 254,879 | 6% | −69,900 | −1% | −27,595 | −2% | 40,611 | 1% | 69,398 | −2% | 3159 | 0% | −9718 | 1% | −52,853 | −1% | 217,559 | 5% | 72,351 | 1% |
| Northern Hungary | −35,257 | −1% | 248,153 | 25% | 77,555 | 2% | −172,099 | −3% | 44,155 | 4% | 612,241 | 10% | 244,386 | −8% | 445,932 | −41% | 696,424 | −84% | 689,578 | 14% | 456,854 | 10% | 616,763 | 9% |
| Norther Great Plain | 246,049 | 5% | 231,875 | 24% | 508,512 | 13% | −329,907 | −5% | −811,480 | −69% | 2,191,988 | 35% | −267,101 | 8% | 201,017 | −19% | 177,892 | −21% | 255,095 | 5% | 253,304 | 5% | 2,346,869 | 34% |
| Southern Great Plain | 92,307 | 2% | 54,688 | 6% | −13,404 | 0% | 243,996 | 4% | 124,237 | 10% | 116,431 | 2% | −56,440 | 2% | 489,875 | −45% | −387,864 | 47% | 125,124 | 3% | 155,814 | 3% | 177,347 | 3% |
| Not allocated | 33,130 | 1% | 110,648 | 11% | 76,184 | 2% | 35,240 | 1% | 90,897 | 8% | 247,919 | 4% | 420,395 | −13% | 445,189 | −41% | 563,645 | −68% | 724,373 | 15% | 755,297 | 16% | 384,988 | 6% |
| **TOTAL** (inc. not allocated) | 5,123,406 | 100% | 974,536 | 100% | 4,038,953 | 100% | 6,152,611 | 100% | 1,183,957 | 100% | 6,197,475 | 100% | −3,241,383 | 100% | −1,086,067 | 100% | −832,753 | 100% | 4,774,979 | 100% | 4,669,791 | 100% | 6,964,105 | 100% |

Note: extreme values are highlighted in orange (less than −100%); yellow (between −100% and −50%); green (between 50% and 100%); blue (more than 100%).

As a personal observation, in one of the towns located in the countryside, major road and railway constructions were carried out by a German multinational company to ensure a faster, smoother delivery of its operational supplies and final products from and to the market. The roads are now in public use too. In another city, a German-speaking kindergarten was founded for the children of foreign expats; its doors are now open for nationals as well. These instances exemplify how not only the infrastructure but also the educational sector might adjust to foreigners' long-term presence.

*4.3. Migration and Human Capital*

After an extensive analysis backed by United Nations (UN) data, Mahtta et al. [37] proved that urbanization is substantially bolstered by population growth and moderately by economic growth and decent governance. The correlations and the volatility of the growth differed across continents and development stages of countries, but were valid in CEE; this statement is validated by the relative contribution of the GDP growth against other relevant factors in contemporary urbanization. The paramount linkages behind these relationships are the investments into production and commercialization, as corroborated by the absorption of the increasing and sophisticating demand, which in turn demands a properly qualified labor force. In the case of Europe, fulfilling this demand was impossible without supplementary workforce immigration, even if only temporarily (conversely, in Africa or Asia, natural population growth covered any potential shortages.) Due to many factors (e.g., geographical proximity to high-income countries, membership of international organization, sea access), growth was a relatively sudden and fast-paced process in Europe, while it is still lingering in many underdeveloped or developing regions worldwide. However, the correlation between urban sprawl and GDP growth has been decreasing with time. No evidence was found regarding correlating THE speed of the two processes [38]; in other words, it depends on the extent to which urbanization reinforces economic growth. Apart from the variable sensitivity, this is in line with the observations in this study. Accordingly, a high population density generates turnover and consumption and, thus, motivates supply and enhances efficiency and productivity—these items all culminate in economic attractiveness. This explains why capital and knowledge are only concentrated in some areas. Additionally, Taubenböck [2] considered density a measure of urbanization.

As a personal input, we would like to draw attention to the 2018 FIFA World Cup, which took place in Russia. During an interview with a foreign emissary, it became apparent that, without people with significant experience in hosting an international event at the local government's disposal, the given country relies on the expertise of multinational companies (e.g., the Portuguese Brandia Central, which was commissioned for the branding of the event); moreover, the country invites a huge number of foreign expats, if not to manage, then at least to supervise the operation and assist in setting the quality standards.

The know-how accumulation capacity of the core is far more advanced than that of CEE. Brain drain is a widely acknowledged phenomenon wherein thousands of graduates migrate to Western Europe in search of superior income prospects [39]. In the meantime, the CEE states attract a workforce from Asia [40]. Notably, immigrants' willingness to integrate is largely contingent on their origins and individual characteristics (e.g., educational attainment), not only on their destinations [39]. As far as origin is concerned, European Christian migrants present a high willingness to assimilate, while people of other religions and cultures may not [38].

CEE's compound net migratory balance has been positive in the past five years [41]. The majority of its non-European population originates from Asia, primarily from China, which accounts for a reinforcing pattern [40]. In contrast, the core hosts mainly European migrants [39], except for asylum seekers. Table 4 shows the distribution of immigrating citizens in 2020 as per their origins. Excluding Slovakia, the average proportion of non-EU immigrants is 77% for CEE (Source: Eurostat [42]), and in the entire EU, the respective

number is 57%. Since the outbreak of the COVID-19 pandemic, non-EU immigration has dwindled.

**Table 4.** Distribution of immigrants by citizenship, 2020.

| Country | Total | European Union (EU) Citizens | | Non-EU Citizens | |
|---|---|---|---|---|---|
| | 1000 | 1000 | % | 1000 | % |
| Bulgaria | 13.4 | 0.9 | 6.7 | 12.5 | 93.3 |
| The Czech Republic | 59.8 | 17.9 | 29.9 | 41.9 | 70.1 |
| Estonia | 9.8 | 3.4 | 34.7 | 6.4 | 65.3 |
| Hungary | 43.8 | 17.1 | 39.0 | 26.7 | 61.0 |
| Lithuania | 22.3 | 0.9 | 4.0 | 21.4 | 96.0 |
| Latvia | 4.6 | 0.5 | 10.9 | 4.1 | 89.1 |
| Poland | 158.3 | 73.8 | 46.6 | 84.5 | 53.4 |
| Romania | 30.8 | 6.1 | 19.8 | 24.7 | 80.2 |
| Slovenia | 24.8 | 3.1 | 12.5 | 21.7 | 87.5 |
| Slovakia | 2.8 | 2.1 | 75.0 | 0.7 | 25.0 |
| Finland | 23.2 | 6.4 | 27.6 | 16.8 | 72.4 |
| Germany | 580.7 | 302.9 | 52.2 | 277.8 | 47.8 |
| Sweden | 65.5 | 19.1 | 29.2 | 46.4 | 70.8 |

A disadvantage of third-country immigration is that it arises with a more complicated, more costly and longer administrative relocation procedure than its intra-EU alternative. Vaccination compliance has been an additional burden recently, which has been put in place for short-term visits too.

International, especially intercontinental, immigrants in CEE are usually from the upper-middle social class and have outstanding employment prospects, better than those in the core [43], and they prefer capital regions. Except for Poland and Romania, only the capitals have airports, which is explained by the countries' small physical extent. Based on our personal experiences, we conclude that the command of foreign languages of the inhabitants is far better in the capitals, regardless of the countries we have visited, compared with smaller cities or towns, except for cases where settlements are tourism-oriented or proximate to borders. Nevertheless, in the latter case, one encounters the language of the neighboring country, not English.

## 5. Evidence of Concept: Performance Gap

Economic output, performance, growth potential, attractiveness, etc., of urban areas excel compared to those of the countryside almost everywhere in the world, culminating in geographical disparities. As in CEE, socialism, in essence, aimed to create equality among the social layers (vertically) as well as in space (horizontally), diminishing the differences between urban and rural areas. Upon shifting to capitalism, however, activities with larger profit and growth potential arbitrarily move into large business centers, accumulating tangible (e.g., airports) and intangible resources (e.g., universities). FDI inflow (resulting in knowledge inflow, stable employment, competitive wages, better living conditions) targets these urbanized settlements, primarily the capitals, thus reinforcing the divergence in resource concentration. The opposite happens in rural areas, especially in depopulating regions, where there is no suitable labor force, nor market with disposable income for generating consumption and tax revenue.

Thus, peripheral and agricultural areas lose competitiveness against cities and metropolitan areas, which offer more lucrative jobs and leisure activities. Consequently, urban expansion and development widens social and economic gaps; this is not specific

to Europe, but it is observable in Asia as well. The approximate process, although taking place in distinct parts of the world, draws attention to the same underlying economic antecedents and regularities. The asymmetric development in Europe also has historical roots in socialism, as preferential state-led production was concentrated on to exploit economies of scale and improve efficiency [2]. Conclusively, FDI has not diverged but enhanced these patterns and deepened domestic disparities further.

The attractiveness of FDI hosting "frontier" areas [4] (p. 45) is interrelated with migratory processes. According to the Harvard Business Review [40], frontier areas "are characterized by politically manipulated markets, weak legal systems, and either low per capita income or faltering GDP [ . . . ] forecast to grow the fastest over the next five years. [ . . . ] [G]lobal investment in developing these resources will continue to boost income and growth. [ . . . ] [G]rowth in frontier economies depends relatively little on overall global economic trends, and first movers can reap better returns on foreign investments". In Europe, people moved into these economic hubs in the pursuit of better living standards. It is not rare for the average employee compensation to be 40–50% higher than the domestic average in capital regions, as exemplified by Poland or the Czech Republic; however, it does not accurately express purchasing power because, in frontier areas, living costs are slightly higher too [4]. Consequently, domestic polarization spilled over after the opening of the capital market and labor, because FDI inflows were driven by the past and socialist deployment of production and further support factors (factories, transportation nodules, allocation points, administrative hubs).

Figures 2 and 3 highlight those urbanized areas that are indeed of superior economic potential. An element of the above-cited definition of urbanization is the "increased number of people . . . "; therefore, not only are the settlement's characteristics relevant but the comparatively high population density is also important. People of a particular social/demographic segment drive tax revenues and consumption and, thus, profit, growth and innovation too. In this vein, Figure 2 (Source: Eurostat [44]) shows the 2021 population density statistics of Europe (there are no census data available regarding 2020). As covered above, Taubenböck [2] also considered population density as a measurement of urbanization. Based on this, we reflected on more urbanized areas, where more people live and work. It is a logical connection because people signify demands for a wider variety of services that are more common in urban spaces (better public transportation, institutions of higher education, etc.) as well as accounting for labor supply for the stable provision of those perks.

As is shown in Figure 2 (Source: Eurostat map [44]), the "average" level of urbanization—measured via population density—is generally more advanced in the core; the highest is in the Benelux states in particular, and the lowest is in agriculture-reliant areas, such as in NUTS 3 regions of landlocked Italy, Spain and Bulgaria. On the other hand, ports and sea access are a major facilitator of trade. However, it is noteworthy that lowly populated areas are not necessarily underdeveloped (exemplified by the Scandinavian region's low density rates).

There is a positive association between GDP per capita, employment rate and quality in terms of value addition of a particular economic sector and labor migration. At the same time, emigration is triggered by poor job opportunities and unemployment and, therefore, directed toward advancement potential. The presence of a virulent knowledge economy is a proxy measure for urbanization, as explained by the following items [29]:

- It requires a sufficiently developed hard and soft infrastructure.
- It requires highly qualified labor masses.
- It relies on the presence of complementary and solvent economic sectors or individual-level demands.

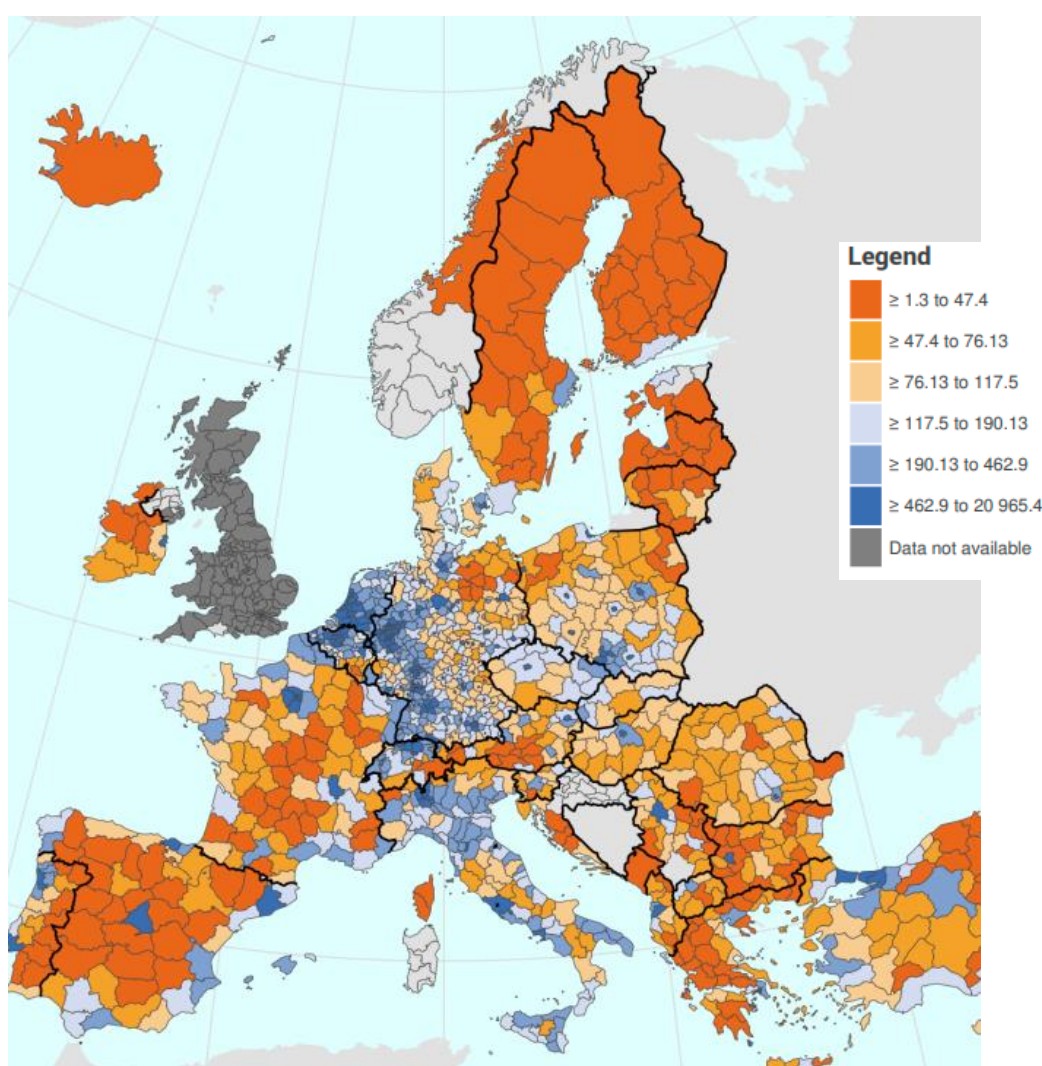

**Figure 2.** Population densities in Europe, NUTS level 3, 2021.

Based on a personal conversation, we can add one more requirement that is not necessarily confined to the knowledge industry but extends to foreign investments in general. The local authorities, legal system and government may result in certain FDIs being considered unfavorable, especially where brownfield investment (acquisition) is concerned, which means that defending domestic ownership may be prioritized over the extra capital. In this case, free market mechanisms are artificially disturbed. An involved person, with whom we had this conversation, had planned to purchase several valuable properties on the Croatian seaside before the completion of the EU accession negotiations (2011). He expected a large profit upon renting or selling them later. However, since he did not trust the reigning legal system and the stability of the government, he did not make this move (e.g., the authors in [45] wrote an article regarding a violent anti-government demonstration in Zagreb:).

Complementary sectors are usually represented by multinational corporations (e.g., car makers, the financial sector, telecommunication companies) that have established themselves in the most urbanized areas. These companies typically survived the 2008 and 2019 downturns, and occasionally experienced growth deriving from the "cheaper" acquisition or skyrocketed demand or share prices (see remote working).

From Figure 3 (Source: ESPON Policy Brief, p. 11. [29]), it can be observed that net emigration characterizes the relatively underdeveloped regions, while immigration and high employment rates in the knowledge economy are observable in Scandinavia,

Western Europe, Northern Italy and Southern England, and most importantly, all capital cities function as domestic hubs. The last point can be foremost exemplified by Romania, Hungary, North-Western France, Lithuania and Bulgaria.

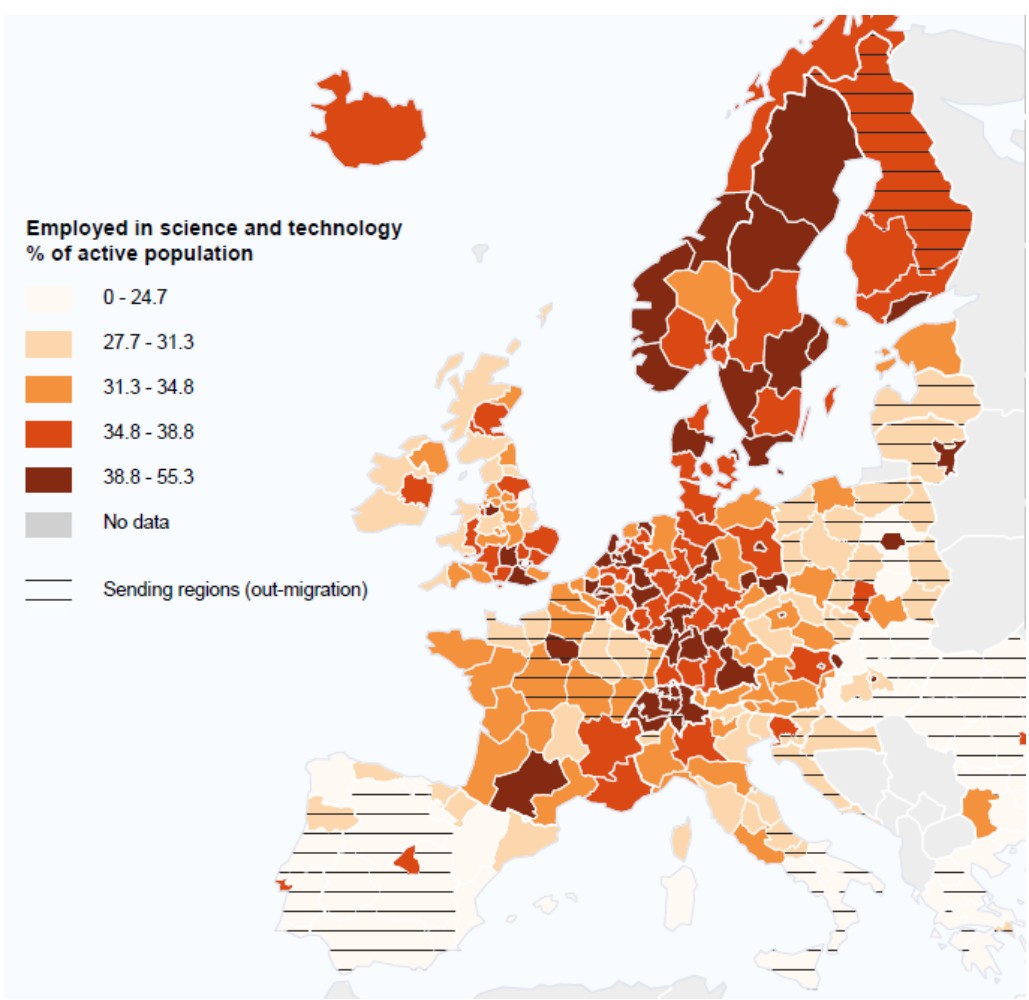

**Figure 3.** The association between knowledge economy and migration, 2019.

Both Figure 2 [44] and Figure 3 [29], based on Eurostat data, illustrate performance gaps between rural and urban areas and between CEE and the core; these gaps correlate with the respective migratory balances and the presence or absence of the knowledge economy of outstanding value-addition capability. Economic advancement and urbanization are therefore enhancing each other. Additionally, we can conclude that, with time, this correlation has strengthened; in this context, we can say that the capitals have always been the engines of economic growth and attractive for investors, both foreign and domestic, and their prosperities are, in certain countries, incomparable to the domestic averages (see Spain, Poland, Lithuania or Romania). Generally speaking, country-wide performance indicators are occasionally, but not always, more balanced in the core countries.

## 6. Discussion

The adoption of capitalism and market competition, the pursuit of economic growth in CEE in the 1990s and the accession to the EU were meant to converge domestic and local economies alike. In contrast, the core underwent a more balanced development over the past century and more recently adopted a paradigm favoring well-being over growth. The relevance of our results lies in the denial of the neoliberal policy, which is a frequently referred to omnipotent principle when it comes to understanding how the

individual CEE countries and the continent as a whole would converge. However, this is not what happens in reality. We investigated how FDI and migration "behave" and which patterns do not comply with the equilibrium: Investments have targeted the inherently developed industrial/urbanized/metropolitan frontier centers, and these host areas improved in terms of their GDP, income levels and employment rates; in contrast, rural and agriculture-reliant areas remained unproductive and unattractive. Domestic and international migration contributed to this widening rural–urban dichotomy: Urban areas saw continuous growth, whereas rural areas lost population. Migrants, especially international migrants, had excellent employment perspectives and brought about further economic advantages (in the form of know-how, tax revenues, consumption, etc.). Accordingly, places with concentrated resources improved at an accelerated pace, while the peripheral areas remained neglected. These phenomena have the tendency to accumulate into self-enhancing centrifugal–centripetal forces, making the diverging paths unlikely to cease by their own mechanisms.

Based on these mechanisms, we defined the following theory: FDI has been motivated or reduced by formerly set conditions, such as the presence of infrastructure, production hubs, natural resources and transportation nodes. The number of influential contextual factors are likely to be uncountable; we did not intend to deliver an exhaustive list in this paper. Labor migration, which is more of a flexible production factor, tends to follow capital movements, because investments, particularly job and employment investments, generate disposable income. The adjustment capability is not confined within the Schengen area; on the contrary, it includes extra-European migration.

According to the neoliberal policy, capital flows where the labor force is cheaper (so that the return on capital is higher), while labor force flows where the capital is relatively abundant but workforce is scarce (so that the demand for workforce pushes the wages up). These opposing powers should reach equilibrium eventually, but in reality, the labor force follows capital, leading to an unequal concentration. In fact, FDI attracts the workforce, both in terms of quality and quantity. The two processes entangle as a vicious circle and do not neutralize/even one another out in the long term; the past three decades have proved that. This vicious circle has been and will be lifting certain areas' economic performance and welfare, while its absence has been damaging and holding back others.

This phenomenon is observable in rural and urban areas as well as in the disparity between CEE and the core. GDP is hardly an indicator in this sense, because it does not reflect on the terms/forms/content/quality of value addition (e.g., knowledge or physical labor) of the local factors (nationals' knowledge, investments), only on the output. This is the reason why we cited knowledge economy instead of gross output.

The present study also has some limitations. The methodology followed is subjective, and the results presented in this paper are confined by the authors' knowledge and the literature reviewed. The two examined processes are undoubtfully explanatory but still insufficient to discover the entire complexity of urbanization. Nevertheless, large geographical units ("CEE" or "capital regions") were handled homogenously or analogously. A very precise quantitative assessment in the future could counterbalance subjectivity, and a more detailed, micro-level scrutiny is also desirable. Research in this area may culminate in policies for a more harmonious balance between urban and rural areas, and Europe-wide development that will eventually achieve convergence.

## 7. Conclusions

The growth of economic divergence in Europe began after World War II (WWII), and the resulting gap widened after its transition and accession to the EU. CEE citizens expected a profound and comprehensive improvement in their standards of living, one that would catch up to the core of Europe, but the convergence still lags behind. The present study analyzed this divergence through urbanization, considering that urbanization is crucial for economic development [16], and we explored two components contributing to the increasing rural–urban economic divergence [1]. First, we studied FDI and (further)

specializations that are partially rooted in the socialist production deployed in urban areas for better efficiency through resource concentration and utility of economies of scale [4]. Second, we explored internal and international migratory processes and found both to be fundamentally oriented toward certain metropolitan hubs.

The research question was as follows: How and why does FDI and labor migration reinforce rural–urban disparities (1) between the core of Europe in CEE and (2) within the individual CEE countries?

1.  As far as rural–urban disparities are concerned, after the economic transition, real experience and know-how regarding market economy and technology arrived with Western capital. FDI inflow was concentrated in inherently more developed frontiers, essentially urbanized areas [4], usually where socialist production hubs were deployed during the previous century. Apart from the quality of the pursued activities, investors of any class preferred frontier urban areas and neglected peripheral rural areas, explained by the latter's poor infrastructure and high out-migration rates threatening a labor shortage. The process bolstered improvements in those areas that, in return, enhanced the divergence among the domestic settlements, and originally underdeveloped (in terms of presence of soft and hard infrastructure, complementary sectors, solvent demand, educational possibilities, etc.) rural areas remained neglected or underutilized. The population grew in preferential areas through immigration (educated labor force and international migration), while the peripheral areas became unattractive because of poor employment prospects [20]. The attractiveness of these areas was driven not only by superior monetary aspects but also by a wider variety of leisure activities, which count on the higher share of disposable income of local residents. Therefore, divergence was reinforced in parallel to internal migration to cities, which was both a domestic and continent-wide phenomenon (see France and Spain).

2.  Regarding the second question, the same two processes are accountable; however, migration especially has been presenting a rather peculiar pattern. Throughout the second half of the last century, CEE cities abruptly became overcrowded, while urbanization showed a more moderate tendency in Western Europe [2]. To build up an independent and diversified domestic economy, a supportive, tenacious institutional background and the culture of the premise of improvement are indispensable. However, CEE misses those components, reflecting the low value addition of both the domestic- and foreign-possessed (FDI) sectors due to their specializations in less-upgraded economic activities, exemplified by fabrication and labor-intensive phases, for instance [13,14]. The knowledge economy is underdeveloped in CEE compared to the core, because of the deployment of the physical labor-intensive activities of the global value chains in the region. This corresponds with emigration of the skilled labor force. They are not deemed "fit" to work in these regions; additionally, they can move flexibly within the EU's free market and pursue better offers in Western Europe or Scandinavia.

Conversely, CEE recruits a non-EU citizen workforce, predominantly from Asia, but that is a more troublesome procedure. However, the advantage of related administrative barriers is that they function as filters. Only those people who are authorized by local employers are allowed to enter; in other words, this form of labor resupply is of distinguished quality (well-educated) and fitting. Such restrictions are not applicable within the Schengen area.

**Author Contributions:** Conceptualization, P.F.; methodology, P.F. and K.G.; writing—original draft preparation, P.F. and K.G.; writing—review and editing, P.F.; visualization, K.G.; supervision, D.F.; project administration, P.F. All authors have read and agreed to the published version of the manuscript.

**Funding:** This research received no external funding and the APC was funded by Széchenyi István University.

**Institutional Review Board Statement:** Not applicable.

**Informed Consent Statement:** Not applicable.

**Data Availability Statement:** Not applicable.

**Acknowledgments:** The authors acknowledge the administrative and technical support.

**Conflicts of Interest:** The authors declare no conflict of interest.

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
