# Peer review of "Elements of Divergence in Urbanization between Central and Eastern Europe (CEE) and the Core of the Continent"

_sustainability, doi:10.3390/su141912377_

Round 1

Reviewer 1 Report (Previous Reviewer 2)

After the improvements are made, the manuscript looks better. Still, in my opinion, it adds nothing new to the knowledge about divergence in urbanization between Central Eastern Europe and the core of the continent. The manuscript consolidates the existing knowledge - an existing theory emerged from the research (in line with the Grounded Theory assumptions).

As the aim of the work, the authors define "Our purpose was to argue why foreign direct investments and labor migration flow are relevant to urbanization and rural-urban disparities and divergences across Europe and in domestic frames (rural-urban areas)". While issues related to migration, population density, etc. are discussed quite widely (supported by statistical data), foreign direct investments are poorly documented (in my opinion it can be illustrated with statistical data on this issue for Europe - NUTS 3 level). In line 66 the authors write "We analyzed capital investments and labor migration ..." But where are the analyzes of capital investments?

Below are also detailed comments:

- line 84-85 - please provide examples of informal and nonscientific sources,

- the title of section 4 Analysis should be clarified - in this approach it is too broad (analisys what?),

- lines 225-249 - Some mistake - should it be in the form of a table?

- lines 267-275 - I'm not sure how this relates to the rest of section 4.1. From state-led socialism to capitalism and liberalism ?,

- lines 322-324 - In my opinion, China is a completely different case of industrialization and transformation. I would expect an appeal to European literature at this point.

- table 2 Immigration by citizenship, 2020 - Remove "%" from the numerical values ​​given in the columns - the unit given in the column header is sufficient

- lines 499-501; lines 545-548 - these conclusions do not result from the data analyzes carried out (see notes above).

Author Response

Comments and Suggestions for Authors

After the improvements are made, the manuscript looks better. Still, in my opinion, it adds nothing new to the knowledge about divergence in urbanization between Central Eastern Europe and the core of the continent. The manuscript consolidates the existing knowledge - an existing theory emerged from the research (in line with the Grounded Theory assumptions).

As the aim of the work, the authors define "Our purpose was to argue why foreign direct investments and labor migration flow are relevant to urbanization and rural-urban disparities and divergences across Europe and in domestic frames (rural-urban areas)". While issues related to migration, population density, etc. are discussed quite widely (supported by statistical data),

We added more statistical data (Table 3) regarding FDI. As far as population density – an outcome of labor migration - is concerned, above Figure 2. we explained why is it appropriate to refer to that when measuring urbanization: “Taubenböck also considered population density as a measurement of urbanization”. So higher density. = more urbanized areas.

foreign direct investments are poorly documented (in my opinion it can be illustrated with statistical data on this issue for Europe - NUTS 3 level).

FDI into NUTS 3 regions were exemplified by Hungary (Table 3.) and the most important message was highlighted. Driven by volume restrictions, no more illustrations were added.

In line 66 the authors write "We analyzed capital investments and labor migration ..." But where are the analyzes of capital investments?

“Foreign” capital we added in the cited sentence as clarification. Decision of focusing foreign, instead of all sources was justified in the first paragraph of section 4.2. We analyzed and proved our statements about FDI concentration through its occupational and geographical (NUTS 3) allotation in Hungary.

Below are also detailed comments:

- line 84-85 - please provide examples of informal and nonscientific sources,

There are examples already:

section 4.1: last paragraph (Financial Times interview cited)

section 4.2: last paragraph (observation)

section 4.3: first paragraph (interview)

section 4.3: last paragraph (experience)

- the title of section 4 Analysis should be clarified - in this approach it is too broad (analisys what?),

More detailed and accurate title added.

- lines 225-249 - Some mistake - should it be in the form of a table?

Clarified up, inserted as Table 1.

- lines 267-275 - I'm not sure how this relates to the rest of section 4.1. From state-led socialism to capitalism and liberalism ?,

Additional clarification, explanation added within the same paragraph.

- lines 322-324 - In my opinion, China is a completely different case of industrialization and transformation. I would expect an appeal to European literature at this point.

Agreed; we found evidences proving the continent-wide variations, see first paragraph of section 4.3. The China-based references and analogies were omitted.

- table 2 Immigration by citizenship, 2020 - Remove "%" from the numerical values ​​given in the columns - the unit given in the column header is sufficient

Done

- lines 499-501; lines 545-548 - these conclusions do not result from the data analyzes carried out (see notes above).

These conclusions are indeed related; see the following simplified explanation

Lines 499-501: “rural vs. preferential areas: neglection vs increase”

Lines 545-548: “resource-concentration”; “self-enhancing procedure”

Section 4.1

socialist production hubs had established and mature infrastructural system and market possibilities right there

already (1990) established resource-concentrating and administrative hubs

Section 4.2

FDI favored those inherently developed spaces, avoiding uncertainty, investments flew there; geographical densification into two NUTS 3 regions

capital, employment and know-how concentration, low chance for unemployment

Section 4.3

better, wider range of job/leisure/education opportunities, better social conditions, e. g. proper transportation system, higher administrative capacities attracted people

people, consumption, knowledge, innovation concentration and social gathering points

Reviewer 2 Report (Previous Reviewer 1)

The article is clearer now. However, I would still recommend an additional round of proofreading before publishing it, as there are still a few phrasing mistakes.

In addition, R226-249 might look better in a table with clear headings. Table 2 might also benefit from editing (in addition, I think that the number separators should be "." (point), instead of "," (comma)).

The quotation in R400-404 has remained written in bold characters, for some reason.

The paragraph between R557 and R564 should be clearer: What does the theory posit? What does it explain and what, if applicable, does it predict? (This is essentially the article's take-home message).

Author Response

Comments and Suggestions for Authors

The article is clearer now. However, I would still recommend an additional round of proofreading before publishing it, as there are still a few phrasing mistakes.

In addition, R226-249 might look better in a table with clear headings.

Done, inserted as Table 1.

Table 2 might also benefit from editing (in addition, I think that the number separators should be "." (point), instead of "," (comma)).

Tables edited; number separators modified and duplicated “%” marks were deleted

The quotation in R400-404 has remained written in bold characters, for some reason.

Recovered to normal characters

The paragraph between R557 and R564 should be clearer: What does the theory posit? What does it explain and what, if applicable, does it predict? (This is essentially the article's take-home message).

We think that we were clear enough in this paragraph but we paraphrased it a bit to make it straighter.

It posits that “the two processes (labor and capital flows) do not neutralize/even out one another, the past three decades had proved that. The last statement here is in perfect progressive tense, predicting a self-reliant process and long duration.

Round 2

Reviewer 1 Report (Previous Reviewer 2)

For figures 2 and 3, please give the source.

Author Response

Dear Reviewer,

Thank you very much for you comments. We've tried to comply with them in our revision the best we could.

For figures 2 and 3, please give the source.

Figure references added

Figure 2: kindly see the paragraphs above and below the figure (Eurostat map, reference no. 44. in this paper)

Figure 3: kindly see the paragraphs above and below the figure (ESPON Policy Brief, p. 11., reference no. 29. in this paper)

We've also added some clarification right below Figure 3.

Please find the corresponding Internet links for these sources in the reference section.

Best regards,

the authors

This manuscript is a resubmission of an earlier submission. The following is a list of the peer review reports and author responses from that submission.

Round 1

Reviewer 1 Report

The article promises to address an important, albeit somewhat neglected, area of enquiry. However, it sometimes fails to deliver at the level promised. I have divided my observations in two categories: A. Major Issues and B. Minor Issues.

A. Major Issues

1. What role does grounded theory play in the article? Where do the authors use it and how do they use it? And furthermore, how strict are they in employing grounded theory (i.e., what is their stance on the "No pre-research literature review" and on the "No talk" rule? Are these rules valid for them, and, if so, how have they been employed?). If grounded theory is central to the authors, I think a short description of how they used it might prove helpful to the reader.

2. How did the authors provide answers to the two highly relevant research questions they posed at the beginning of the article (R105-107)? How does the argumentation reflect in the structure of the paper?

3. Why did the authors choose informal networks/structures, FDI inflow, and migration as relevant arguments for answering the research questions? In other words, why three factors and why these three? In addition, what do the authors exactly mean by "informal networks"?

4. How did the authors choose the relevant thresholds for each of the three factors mentioned above? (see also the point below).

5. I did not quite understand the methodological section: What are the "unique socioeconomic characteristics of CEE" (R59) and how did the authors select them? The text is somewhat unclear in this respect.

6. Without a clear methodological section, the sequence of results in the following section is sometimes difficult to follow: Why does the theoretical background appear in the "Results" section, instead of the introductory section? Why do convergence and divergence and present division [of] pertain to the theoretical background, whereas the "The Legacy of socialism" stands alone?

7. What do the authors exactly mean by the "present division" (R124) and the "urban-rural performance gap" (R262)?

8. How is the discussion section linked to the results section? The discussion section seems to be rather descriptive and historical in character, without going too much into the relevance of the results obtained.

9. Answers to the two research questions should be clearly stated. In this respect, the conclusions sections remains somewhat vague.

B. Minor Issues

1. The text needs an additional round of proofreading and editing, as there are still a few mistakes left.

2. I did not quite understand the title of Figure 1. Is it about the level of urbanisation? (What does "rural-urban" dispersion mean?)

Reviewer 2 Report

The primary remark regarding this manuscript is that it adds nothing new to the understanding of divergence in urbanization between Central Eastern Europe and the core of the continent.

The authors indicated grounded theory (GT) as the research method used - it is a methodology that includes a set of guidelines for collecting and analyzing data in order to construct a theory from them. The grounded theory methodology is based on literature, but the analysis is most often performed at the end of the research procedure, not at its beginning. Here, the entire research procedure is based on the analysis of the literature. Additionally, is the number of references (around 30) not too small with such assumptions?

The authors' own contribution is an analysis of the literature. This is only one of the elements of GT, in this case the assumptions are that the theory will emerge only from the analysis of existing texts - is it a bit too little? And where is the research material from observations, interviews and other sources?

The basic question - what new theory has been constructed as a result of the literature analysis?

Below are also detailed comments:

- abstract - the authors write about the frontier areas of Central Eastern Europe; this is misleading as it concerns the frontier urban area (the same error is reproduced in the further part of the text),

- each time an abbreviation is used for the first time, its meaning must be explained (e.g. the FDI used in the abstract is not explained throughout the manuscript),

- urban transformation of the CEE countries is discussed very briefly,

- the purpose of the research is not explicitly discussed in the Introduction section,

- the Conclusions section should clearly indicate what new research results bring to the state of knowledge (this is a deeper problem related to this manuscript),

- line 69-72 "As the actual progress in embedded in a unique place as framework in every case, the contextual elements (history, culture, economy, social patterns, psychology, politics, perception, formal knowledge etc.) of every location had to be scrutinized [10] ”- is every location has been scrutinized in terms of contextual elements?

- data from different periods are compiled and compared: Figures 2 and 3 - 2013, Table 1 - 2011, Figure 1 - 2020; how do you compare it with each other?

- figure 1 - The colors used in the diagram should be described in the legend, the vertical axis should be described (and not in the legend caption)

- similarly figures 2 and 3 - they are "cited", but they are thematic maps and should be given scale, north direction, legend (and not the note in the description "Darker color indicates higher value"); one more remark - the figures show generally available data - shouldn't it be better to analyze the current data - it falls within the scope of GT,

- table 1 - the signature should include the year,

- twice the authors emphasize that they omitted something because they focused on something else - what is it for? (lines 110-111, 158-160),

- do figures 2 and 3 really visualize urban – rural dichotomies? In my opinion, they show the dichotomy between the regions of the EU. How the reader should know which areas are urban and which are rural (no urban centers are marked).

- lines 412-413 - is it probably a mistake?

- references 31 and 32 - how it relates to the subject of the manuscript - relate to China.

I encourage to improve the manuscript and resubmit.

Reviewer 3 Report

The submitted paper has a number of simplistic views and inaccuracies. Its major weakness is the absence of a detailed literature review - post-socialist (and post-communist) development has been the subject of a number of authors across Europe and beyond. I also find the very short section on methods problematic. Moreover, the authors have chosen only selected aspects to assess the development of the post-socialist countries - these are very diversified from country to country and in some cases cannot be compared (e.g. by different methodologies for calculating unemployment rates, etc.). Similarly, the approach of states to promoting FDI was different in the Eastern Bloc, so it cannot be simply compared in this way. The relationship to urbanisation is then very loose and often purposeful. There are also errors in the text - e.g. the city of Pardubice does not have the automotive industry as a key aspect of its economy, while the name of the city is also incorrectly spelt in the paper). The authors do not respect the local specifics and different regional policy approaches that have been reflected in the selected indicators. It should also be added that the interpretation of the results is rather vague, without context and without link to the main theme of the Special Issue. In view of the above, I do not recommend the article for publication, even after revision.